# Characterization of Lipid Profiles after Dietary Intake of Polyunsaturated Fatty Acids Using Integrated Untargeted and Targeted Lipidomics

**DOI:** 10.3390/metabo9100241

**Published:** 2019-10-21

**Authors:** Satoko Naoe, Hiroshi Tsugawa, Mikiko Takahashi, Kazutaka Ikeda, Makoto Arita

**Affiliations:** 1Laboratory for Metabolomics, RIKEN Center for Integrative Medical Sciences, Yokohama 230-0045, Japan; satoko.naoe@mochida.co.jp (S.N.); kazutaka.ikeda@riken.jp (K.I.); 2Metabolome informatics research team, RIKEN Center for Sustainable Resource Science, Yokohama 230-0045, Japan; mikiko.takahashi@riken.jp; 3Cellular and Molecular Epigenetics Laboratory, Graduate School of Medical Life Science, Yokohama City University, Tsurumi, Yokohama 230-0045, Japan; 4Division of Physiological Chemistry and Metabolism, Graduate School of Pharmaceutical Sciences, Keio University, Minato-ku, Tokyo 105-8512, Japan

**Keywords:** arachidonic acid, omega-3 fatty acids, lipidomics, mass spectrometry, dietary fat, fatty acid metabolism

## Abstract

Illuminating the comprehensive lipid profiles after dietary supplementation of polyunsaturated fatty acids (PUFAs) is crucial to revealing the tissue distribution of PUFAs in living organisms, as well as to providing novel insights into lipid metabolism. Here, we performed lipidomic analyses on mouse plasma and nine tissues, including the liver, kidney, brain, white adipose, heart, lung, small intestine, skeletal muscle, and spleen, with the dietary intake conditions of arachidonic acid (ARA), eicosapentaenoic acid (EPA), and docosahexaenoic acid (DHA) as the ethyl ester form. We incorporated targeted and untargeted approaches for profiling oxylipins and complex lipids such as glycerol (phospho) lipids, sphingolipids, and sterols, respectively, which led to the characterization of 1026 lipid molecules from the mouse tissues. The lipidomic analysis indicated that the intake of PUFAs strongly impacted the lipid profiles of metabolic organs such as the liver and kidney, while causing less impact on the brain. Moreover, we revealed a unique lipid modulation in most tissues, where phospholipids containing linoleic acid were significantly decreased in mice on the ARA-supplemented diet, and bis(monoacylglycero)phosphate (BMP) selectively incorporated DHA over ARA and EPA. We comprehensively studied the lipid profiles after dietary intake of PUFAs, which gives insight into lipid metabolism and nutrition research on PUFA supplementation.

## 1. Introduction

Polyunsaturated fatty acids (PUFAs) are essential nutrients that have a range of biological effects such as on brain function, cardiovascular disease, obesity, cancer, and bone health in humans [1,2], and also affect the reproduction quality of livestock animals [3]. Among PUFAs, arachidonic acid (ARA), eicosapentaenoic acid (EPA), and docosahexaenoic acid (DHA) are known as long chain PUFAs, and their physicochemical properties and metabolisms provide various functions in mammalian cells. ARA and EPA/DHA are also known as ω6 and ω3 polyunsaturated fatty acids, respectively, and their biological functions and the importance of ω3/ω6 fatty acid balance in maintaining homeostasis have attracted increasing attentions [1,2]. Mammals cannot synthesize ARA, EPA, and DHA from their precursors in sufficient quantities, i.e., linoleic acid (LA) and alpha-linoleic acid (ALA), and they must rely on dietary intake. Therefore, the dietary intake of ARA, EPA, and DHA is essential to maintain human health, and it is well known that livestock animals and marine foods are the primary nutrient sources for ARA and EPA/DHA, respectively.

The lipid profiles in tissues are affected by different balances of dietary PUFAs [4,5,6,7,8,9]. In fact, the intake of ω3 PUFAs increases tissue levels of ω3 PUFAs and their oxidized forms, namely oxylipins, while the amount of ω6 PUFAs and their oxidized forms is decreased [4,5]. Moreover, a previous study revealed that dietary intake of ω3 PUFA inhibits ARA biosynthesis from LA by the suppression of fatty acid desaturase 2 (FADS2) expression [6], and competes with ARA for membrane phospholipid remodeling. In addition, several ω3 PUFA intervention studies have shown that the supplementation of ω3 PUFA increases ω3-PUFA-derived oxylipins while decreasing ARA-derived oxylipins in human peripheral blood [7]. Although oxylipin profiles under different dietary conditions have been reported [4,5,8,9], there have been few studies that comprehensively investigated the lipid profile after the dietary intake of ARA, EPA, or DHA side-by-side in various tissues, which would be necessary to grasp the effects of different PUFA-containing diets. 

After dietary intake, PUFAs are actively incorporated into cells as acyl chains of membrane phospholipids and other lipid classes, such as triacylglycerol and cholesteryl esters. Intake of ARA, EPA, and DHA affects several physiological functions, such as membrane scaffold formation to harmonize biomolecule interactions [10], energy storage and related functions to maintain metabolism [11], and signal transduction via bioactive lipid mediators produced by cyclooxygenases (COX), lipoxygenases (LOX), and cytochrome P450 (CYP) [12]. Since these functions are accomplished by a molecular diversity of lipid species, the comprehensive profiling of lipids in cells and tissues is important to understanding the mechanisms underlying the effect of different PUFA supplementations. 

Liquid chromatography coupled with high resolution tandem mass spectrometry (LC-HR-MS/MS) and triple quadruple mass spectrometry (LC-QqQ/MS) are popular techniques for untargeted and targeted lipidomic analysis, respectively [13,14]. LC-QqQ/MS-based targeted lipidomics has high sensitivity, and has often been used for the profiling of oxylipins, where the concentrations in plasma range from 10 pM to 100 nM [14]. In contrast, LC-HR-MS/MS, which has high resolution and scanning speed, has often been used for the profiling of glycerol (phospho) lipids, sphingolipids, and sterols, where concentrations in plasma range from 10 nM to 100 µM [14]. The current informatics technique for untargeted lipidomics enabled us to characterize more than 90 lipid classes by untangling the MS/MS spectrum [15]. Mass-spectrometry-based lipidomics, therefore, has the potential to comprehensively examine tissue lipid profiles in detail, and this lipidomic profiling could give us new insights into dietary PUFA distribution and metabolism in the body. 

In this study, we investigated the lipid profiles in plasma and tissues including the liver, kidney, white adipose, skeletal muscle, heart, small intestine, lung, brain, and spleen after dietary intake of the ethyl ester form of ARA (ARA-E), EPA (EPA-E) or DHA (DHA-E) in mice. In total, 1026 molecular species of lipid were identified, of which 915 and 111 were annotated in untargeted and targeted analyses, respectively. 

## 2. Materials and Methods

### 2.1. Standard Chemicals

ARA-d8, 15-HETE-d8, LTB_4_-d4 and PGE_2_-d4 were obtained from Cayman Chemical (Ann Arbor, MI, USA). Highly purified EPA ethyl ester (EPA-E) (> 98%), DHA ethyl ester (DHA-E) (> 97%), and ethyl arachidonate (ARA-E) (> 99%) were obtained from Nippon Suisan Kaisha, Ltd. (Tokyo, Japan), Harima foods, Inc. (Osaka, Japan) and NuChek Prep, Inc. (Elysian, MN, USA), respectively. All solvents of LC/MS grade were obtained from Wako (Tokyo, Japan).

### 2.2. Animals

Male C57BL/6J mice (Japan SLC, Inc., Shizuoka, Japan) were purchased at 10 weeks of age, and housed under controlled temperature and lighting (12 h light/dark cycle) with free access to water and a controlled diet (fish-meal-free F1, Funabashi Farm, Chiba, Japan). The fatty acid composition and other nutritional factors for fish-meal-free F1 are described in Appendix A. In this study, the fish-meal-free F1 was used as an ARA-, EPA-, and DHA-free diet, and each PUFA was supplemented into the feed in its ethyl ester form. After 1 week of acclimation, mice were assigned to four groups (n = 5) and fed control diet (control group), control diet supplemented with 1% EPA-E (*w/w*) (EPA-fed group), 1% DHA-E (*w/w*) (DHA-fed group), or 1% ARA-E (*w/w*) (ARA-fed group) for 2 weeks. Animals were dissected under isoflurane anesthesia, after which organs, such as the liver, kidney, white adipose, skeletal muscle, heart, small intestine, lung, brain, and spleen, were collected and plasma was isolated. For the lipidome analyses, four biological replicates (n = 4) were used for liver tissue, and five biological replicates (n = 5) were used for the other tissues and plasma. Two of the white adipose samples from the EPA-fed group and one of the skeletal muscle samples from the ARA-fed group in were excluded from the targeted lipidomics results owing to LC-MS analysis failure. All animal experiments were carried out in accordance with the guidelines for the use and care of laboratory animals of Mochida Pharmaceutical, and were approved by the Institutional Animal Care and Use Committee (identification code: PMS14-041).

### 2.3. Determination of Plasma Triglyceride, Cholesterol, and Fatty Acid Composition

Standard enzymatic methods were used to determine plasma total cholesterol (TC) and triglyceride (TAG) with commercially available kits purchased from Wako Pure Chemical Industries Ltd. Total fatty acids in plasma were analyzed by gas chromatography (Japan SLC, Inc., Shizuoka, Japan). C16:0, C16:1, C18:0, C18:1, C18:2, C20:3, C20:4, C20:5, C22:5, and C22:6 levels were determined using the standard substances: palmitic acid, palmitoleic acid, stearic acid, oleic acid, linoleic acid, dihomo-γ-linoleic acid, arachidonic acid, eicosapentaenoic acid, docosapentaenoic acid, and docosahexaenoic acid, respectively.

### 2.4. Untargeted LC-MS/MS-Based Lipidomics

Total lipids from organs were extracted as previously described [16] using internal standards: 1 µM TAG (8:0/8:0/8:0)-^13^C3 (Larodan, Inc., Monroe, MI, USA), 0.5 µM phosphatidyl glycerol (17:0/14:1) (Avanti Polar Lipids, Inc., Alabaster, AL, USA), 0.5 µM lyso phosphatidylcholine (17:1) (Avanti Polar Lipids, Inc.), 5 µM acylcarnitine (18:0)-d3 (Larodan, Inc., Monroe, MI, USA), 25 µM palmitic acid-d3 (Olbracht Serdary Research Laboratories, Toronto, ON, Canada), 25 µM stearic acid-d3 (Olbracht Serdary Research Laboratories, Toronto, ON, Canada), 5 µM cholic acid-d4 (Cambridge Isotope Laboratories, Inc., Tewksbury, MA, USA), and 0.5 µM Cer/Sph mixture I (Avanti Polar Lipids, Inc., Alabaster, AL, USA) (Appendix A). Untargeted analysis was performed using an ACQUITY UPLC system (Waters, Milford, MA, USA) coupled with a quadruple time-of-flight/MS (TripleTOF 5600+, SCIEX, Framingham, MA, USA). LC separation was performed using a reverse-phase column (Acquity UPLC BEH peptide C18; 2.1 × 50 mm, 1.7 µm particle size; Waters, Milford, MA, USA) with a gradient elution of mobile phase A (methanol: acetonitrile: water = 1:1:3, *v/v/v* for volume ratio containing 5 mM ammonium acetate and 10 nM EDTA) and mobile phase B (100% isopropanol containing 5 mM ammonium acetate and 10 nM EDTA), and the composition was produced by mixing those solvents. LC gradient consisted of holding solvent (A/B: 100/0) for 1 min, then linearly converting to solvent (A/B: 60/40) for 4 min, linearly converting to solvent (A/B:36/64) for 2.5 min and holding for 4.5 min, then linearly converting to solvent (A/B: 17.5/82.5) for 0.5 min, linearly converting to solvent (A/B: 15/85) for 8.5 min, and linearly converting to solvent (A/B: 5/95) for 1 min followed by returning to solvent (A/B: 100/0) and holding for 5 min for re-equilibration. The flow rate and column temperature were set to 0.3 mL/min and 45 °C, respectively. Data dependent MS/MS acquisition mode was applied as previously described [16]. Briefly, the temperature and ion spray voltage floating were set to 300 °C and −5.5 kV, respectively. The accumulation times for MS1 and MS/MS were set to 100 ms and 50 ms, respectively, for scanning a mass range from *m/z* 75 to *m/z* 1250. The collision energy (CE) was set to 35 eV with a CE spread of 15 eV in high-resolution mode, and other settings of DDA mode were as follows: 10 most intense ions, 100 cps intensity threshold, and 100 sec exclusion time.

### 2.5. Targeted LC-MS/MS-Based Lipidomics

The LC-MS/MS analysis was performed as described previously [12]. Samples were extracted by solid-phase extraction using Sep-Pak Vac 3cc C18 cartridges (Waters) with deuterium-labeled internal standards at a final concentration of 10 pg/µL ARA-d8, 10 pg/µL 15-HETE-d8, 10 pg/µL LTB_4_-d4, 10 pg/µL PGE_2_-d4 per samples. The targeted analysis was performed using a UPLC system (Waters UPLC, Waters, Milford, MA, USA) with a triple quadruple linear ion trap mass spectrometer (QTRAP 5500; SCIEX, Framingham, MA, USA), equipped with Acquity UPLC BEH C18 column (1.0 × 150 mm, 1.7 µm particle size; Waters, Milford, MA, USA). Samples were eluted with a mobile phase composed of water/acetate (100:0.1, *v/v*) and acetonitrile/methanol (4:1, *v/v*) (73:27) for 5 min, and ramped to 30:70 over 15 min, to 20:80 over 25 min and held for 8 min, ramped to 0:100 over 35 min, and held for 10 min with flow rates of 70 µL/min (0–30 min), 80 µL/min (30–33 min), and 100 µL/min (33–45 min). The MS/MS analyses were performed in negative ion mode, and oxylipins were identified and quantified by multiple reaction monitoring (Appendix A). Compounds were quantified by using stable internal standards. The calibration curves of all compounds were acquired in triplicate (Appendix A). The extraction and matrix recovery were calculated with internal standards (IS) and the compound quantification was conducted with the calibration curves. The lipid compounds of (group 1) free fatty acids, (group 2) monohydrides and epoxides, (group 3) diols, leukotrienes, thromboxanes, lipoxines, and resolvins, and (group 4) prostaglandins and others were corrected with ARA-d8, 15-HETE-d8, LTB_4_-d4, and PGE_2_-d4, respectively. The compound concentrations were calculated as:$$\mathrm{Compound}\;\mathrm{concentration}=\frac{\left(\mathrm{target}\;\mathrm{compound}\;\mathrm{peak}\;\mathrm{area}\right)/\left(\mathrm{slope}\;\mathrm{of}\;\mathrm{calibration}\;\mathrm{curve}\right)}{\left(\mathrm{IS}\;\mathrm{peak}\;\mathrm{area}\right)/\left(\mathrm{IS}\;\mathrm{slope}\;\mathrm{of}\;\mathrm{calibration}\;\mathrm{curve}\right)}$$

### 2.6. Data Analysis

The targeted analysis data was analyzed using MultiQuant software (SCIEX, Framingham, MA, USA). The chromatogram peaks were manually curated. The untargeted analysis data was analyzed using MS-DIAL software [17,18] version 2.90 (http://prime.psc.riken.jp/) with the following parameters: retention time start: 0 min; retention time end: 100 min; mass range start: 0 Da; mass range end: 5000 Da; accurate mass tolerance (MS1) tolerance: 0.01 Da; MS2 tolerance: 0.025 Da; maximum charge number: two; smoothing method: linear weighted moving average; smoothing level: 3; minimum peak width: five scans; minimum peak height: 1000; mass slice width: 0.1 Da; sigma window value: 0.5; MS2Dec amplitude cut off: 0; exclude after precursor: true; keep isotope until: 0.5 Da; keep original precursor isotopes: false; exclude after precursor: true; retention time tolerance for identification: 10 min; MS1 for identification: 0.01 Da; accurate mass tolerance (MS2) for identification: 0.05 Da; identification score cut off: 80%; using retention time for scoring: true; relative abundance cut off: 0; top candidate report: true; retention time tolerance for alignment: 0.1 min; MS1 tolerance for alignment: 0.015 Da; peak count filter: 0; adduct ion setting: [M+H]^+^, [M+NH_4_]^+^, [M+Na]^+^, [M+C_2_H_3_N+H]^+^, [M+H-H_2_O]^+^, [M+C_3_H_8_O+H]^+^, [M+C_2_H_3_N +Na]^+^, [M-C_6_H_10_O_5_+H]^+^, [2M+H]^+^, [2M+NH_4_]^+^, [2M+Na]^+^ in positive ion mode and [M-H]^−^, [M-H_2_O-H]^−^, [M+CH_3_COO]^−^, [M-C_6_H_10_O_5_-H]^−^, [2M-H]^−^, and [2M+CH_3_COO]^−^ in negative ion mode. All lipid classes supported in the MS-DIAL version were used for the targeted lipids. The annotation was performed using the in silico spectral library of lipids, and the false positive annotations were manually curated by mass spectrometry experts. The principal component analysis and circus plots were executed in the R language with the packages of “prcomp” and “OmicCircos”. Our untargeted lipidomic data were discussed in terms of relative quantification, defined by lipidomic standard initiative (https://lipidomics-standards-initiative.org/), among dietary conditions in each tissue. All of the mass spectrometry data are available at RIKEN DROP Met (http://prime.psc.riken.jp/menta.cgi/prime/drop_index, ID= DM0030).

## 3. Results

### 3.1. Lipid Profiles after Dietary Intake of ARA, EPA, or DHA

We first examined the profiles of total cholesterol, triacylglycerol (TAG), and total fatty acids in mouse plasma. Dietary intake of 1% (*w/w*) ARA-EE, EPA-EE, or DHA-EE for 2 weeks significantly decreased plasma cholesterol and decreased triacylglycerol levels as compared to the levels of the control group (Figure 1a,b). There were no body weight loss, suggesting that PUFA supplementation did not affect the amount of dietary consumption (Figure 1c). Total fatty acid levels, including C16:0 (palmitic acid), C18:1, and C18:2, were decreased after dietary intake of PUFAs, while C20:4 (ARA), C20:5 (EPA), and C22:6 (DHA) were increased in the respective dietary conditions (Figure 1d). Here, abbreviations of C18:1 and C18:2 were used because of the unclear chromatogram separation from several isomers including vaccenic acid (18:1 n-7), oleic acid (18:1 n-6), linoleic acid (n-6), and conjugated linoleic acids. In this study, we fed PUFAs in synthetic ethyl ester form (PUFA-EE) because this method of producing ethyl esters can achieve a high degree of fatty acid purity. However, the bioavailability of the ethyl ester forms has been reported to be lower than the TAG form, which is the major form of PUFAs in natural fish oil [19,20]. Dietary PUFA-EE and TAG are both hydrolyzed to free fatty acid by pancreatic lipase and passively transported into enterocytes. In the enterocytes, free fatty acids are re-esterified to TAG and then transferred into lymph and blood circulation. Among those processes, the digestion of the ester bond and re-esterification may cause the difference of bioavailability [21]. However, the plasma PUFA levels were increased as expected after dietary intake of PUFA-EEs (Figure 1d), supporting the validity of this study design.

Next, we performed untargeted and targeted lipidomic analysis on the plasma and the nine tissues. A total of 915 molecules from 33 lipid classes were characterized in the untargeted analysis, while 111 oxylipins derived from LA, ALA, ARA, EPA, and DHA were identified in the targeted analysis (Table 1) The score plots of the principal component analysis (PCA) from untargeted lipidomics were clearly clustered by dietary-fed group in the metabolic organs, such as the liver and kidney, and the factor loadings showed that the profiles of glycerolipids and glycerophospholipids contributed to the principal components (Figure 2a and Appendix A). On the other hand, the lipidome in the brain and muscle tissues, including the heart and skeletal muscle, was not significantly changed between different dietary conditions. Interestingly, the score plots of DHA dietary intake were classified independently from other conditions in the skeletal muscle, adipose, and heart. According to the factor loadings, the profile of DHA-containing lipids substantially contributed to these score plot dimensions. In fact, the muscle tissues were enriched in DHA present in membrane phospholipids [22], and our results suggested that the efficiency of DHA incorporation into membrane phospholipids was higher in muscle tissues compared to other tissues, and that the PC, PE, and PI lipid classes highly incorporated DHA as an acyl chain component in both the skeletal muscle and heart. In contrast, the membrane phospholipids incorporating DHA and the score plots of dietary DHA intake were not significantly changed in the brain.

### 3.2. Oxylipin Profiles in Targeted Lipidomics

According to the PCA score plots of the targeted lipidomic analysis, the oxylipins and their related molecules of plasma and tissues, except for the brain, were significantly affected by the dietary intake of PUFAs (Figure 2b and Appendix A). Further detail of the oxylipin profiles is given in Figure 3. In most organs, except for the small intestine, lung, and spleen, the concentrations of 11,12-epoxyeicosatrienoic acid (EET) and 14,15-EET were higher than those of other oxylipins in ARA-derived metabolites, including epoxides, hydroxides, prostaglandins (PGs), and others. In contrast, COX-derived PGs such as PGE_2_, PGD_2_, and 6-keto-PGF_1α_ were enriched in the small intestine, lung, and spleen, and these tissues contained higher amounts of hydroxyeicosatetraenoic acids (HETEs). 

The levels of EPA and EPA-derived oxylipins were in fact lower than those of ARA and DHA in plasma and all tissues (Figure 1d and Appendix A), while their amounts were substantially increased by dietary EPA intake as indicated by EPA-derived metabolites (Figure 3). Among the EPA-derived oxylipins, epoxyeicosatetraenoic acids (EpETEs) were the primary oxylipins present in the liver, kidney, white adipose tissue, skeletal muscle, heart, lung, and brain, while PGs and hydroxyeicosapentaenoic acids (HEPEs) were enriched in the small intestine and spleen. A similar pattern was observed in dietary DHA intake; epoxydocosapentaenoic acid (EpDPE) and hydroxydocosahexaenoic acid (HDoHE) were characteristically present in tissues.

The ARA-, EPA-, and DHA-derived epoxides, which are known as CYP metabolites, were enriched in metabolic tissues such as the liver, kidney, and white adipose tissue. ARA-derived epoxides activate various signaling pathways, and elicit functional responses such as vasorelaxation and anti-inflammation [23]. EPA-derived epoxides have anti-allergic and anti-inflammatory properties: 17,18-EpETE inhibits mast cell degranulation [24,25] and 12-hydroxy-17,18-EpETE suppresses neutrophil infiltration and eosinophilic inflammation [26,27]. These epoxides are converted into diols by soluble epoxide hydrolase, and the levels of diols and the diols/epoxide ratio were high in the small intestine (Appendix A). 

The ω3 oxylipins of EPA were increased following the dietary intake of DHA, while the amounts of ARA-derived oxylipins were decreased by ω3 PUFA dietary intake. Similar observations have also been reported in several other studies [5,7,28]. Interestingly, the increase in brain EPA levels following DHA intake was relatively high, and this result may represent the retro-conversion from DHA to EPA [29]. In contrast, ARA metabolism was competitively suppressed by a rich intake of EPA and DHA. 

### 3.3. Lipidomic Signatures in Untargeted Analyses

We used circus plots to display the result of untargeted lipidomics (Figure 4) for the liver and brain, the metabolites of which were the most and least affected by the dietary intakes of PUFAs, respectively (Figure 2a). The heatmap layer and correlation linkages in the circus plot clearly showed that the PUFAs in liver were incorporated into glycerolipids, glycerophospholipids, and cholesterol esters, while the profile of sphingolipids was not affected by any dietary supplement. Moreover, the heatmap layer and the detail of fatty acid compositions indicating the existence of 18:2, 20:4/22:4, 20:5/22:5, and 22:6 as the acyl chain composition in a certain lipid class showed that glycerol (phospho) lipids containing 22:4 and 22:5 were increased in most tissues after the dietary intake of ARA and EPA, respectively. This result indicated that ARA and EPA were elongated to docosatetraenoic acid (DTA, 22:4) and docosapentaenoic acid (DPA, 22:5) by elongases such as elongation of very long chain fatty acids protein (ELOVL) 2 and 5 [30]. In contrast, an increase in lipids containing the elongated product of DHA, i.e., 24:6, was not observed following DHA dietary supplementation in the tissues examined. 

A few differences in lipid profiles were observed in the mouse brain: free fatty acids (FA) of ARA, EPA, and DHA were affected by dietary intake (Figure 4). Moreover, several phospholipids, including phosphatidylcholine (PC), phosphatidylethanolamine (PE), phosphatidylserine (PS), and ether linked PE (EtherPE), partially reflected the changes in dietary conditions according to the heatmap layer. Interestingly, the amounts of most lipid classes, including sphingolipids, were increased in the ARA-supplemented mice. Moreover, untargeted lipidomics identified the mono-galactocyl/glycosyl-diacylglycerol (MGDG) lipids, the main component of plant membrane lipids, in the mouse brain, where the MGDG lipids were composed of saturated or monounsaturated fatty acids. A few studies have also shown the existence of MGDG in the brain, and its biosynthesis and physiological role have been examined in several in vitro studies, although their mechanisms are still unclear [31,32,33]. These results suggest that the annotation of untargeted lipidomics should be performed first in an unbiased manner. 

Untargeted lipidomics also revealed unique lipid features in the other tissues. The diacylglycerol (DAG) and TAG lipids were the most abundant species in the white adipose tissue, as has previously been described [34], and the DAG and TAG profiles were significantly affected in most tissues. Interestingly, however, we found that the profiles of DAG and TAG were not affected in the spleen, although the profiles of free fatty acids, glycerophospholipids, and cholesteryl esters did reflect the intakes of ARA, EPA, and DHA (Appendix A). This result suggested that the metabolic pathway has a limited ability to incorporate dietary PUFAs into TAG in the spleen. Indeed, diglyceride acyltransferase (DGAT) activity has been reported to be low in the spleen [35]. Our results also showed that the ether-linked PC and PE (EtherPC and EtherPE) incorporating *O*-alkenyl or *O*-alkyl chains were enriched in the kidney, spleen, small intestine, muscle, heart, and brain (Table 1), while the acyl chain profiles of EtherPC and EtherPE reflected the dietary PUFA in all tissues and plasma. In contrast, the low variety of *O*-alkenyl or *O*-alkyl chains containing PC and PE in the liver can be explained by the low expression of the alkyldihydroxyacetonephosphate synthase (AGPS) [36].

In addition, we found that the levels of phospholipids containing LA (LA-PLs) were substantially decreased after the dietary intake of ARA (Figure 5a), and profound changes were observed in PC and PE. This may have resulted from competition for the PL remodeling enzyme lysophospholipid acyltransferase 3 (LPCAT3), which has a high affinity for both ARA and LA [37]. Since free LA was decreased by dietary intake of ARA, in addition to LA-PLs (Appendix A), the decrease of LA under ARA supplementation could reflect a potential mechanism whereby total ω6 fatty acid amount, i.e., LA + ARA, is maintained in mammalian cells: in fact, a previous study provided a complementary report in which an ARA decrease is observed in plasma following a higher intake of LA [38]. We also found that the BMP containing DHA (DHA-BMP) was substantially increased under DHA supplementation, while an increase in BMP containing ARA and EPA was not observed in ARA or EPA supplementation (Figure 5b): the annotation was performed by curating both positive and negative ion MS/MS spectra (Figure 5c). While DHA is known to be the predominant fatty acyl chain of BMP in the brain [39], our study suggested that there is a selective mechanism in which DHA is preferably incorporated into BMP in several tissues, including the small intestine, lung, heart, kidney, spleen, and liver. Although the BMP biosynthesis mechanism is still unclear, one study reported that lysocardiolipin acyltransferase 8 (AGPAT8), preferring unsaturated fatty acids as the acyl donor, catalyzed the acylation of BMP [40,41], and the gene expression level of AGPAT8 was high in the small intestine, heart, liver, and kidney in the BioGPS dataset (http://ds.biogps.org/?dataset=GSE10246&gene=225010). Importantly, these results indicate the importance of untargeted lipidomics, where unexpected and novel insights can be discovered in a data-driven manner.

## 4. Discussion

In this study, we applied lipidomics techniques to monitor the comprehensive lipid profiles of mouse plasma and tissues, and their changes following the dietary intake of different PUFAs: ARA, EPA, and DHA. Importantly, we examined the effect of dietary supplementation of pure ARA, EPA, or DHA instead of dietary oils such as fish, linseed, and sunflower oils, and investigated the plasma and nine organs to better understand the common and/or differential lipid modulations in tissues. Mouse tissues, except for the brain, effectively incorporated the dietary PUFAs into glycerolipids and glycerophospholipids, and tissue levels of free fatty acids and oxylipins were well correlated with dietary PUFA intakes. It should be noted that our lipidomics data provided the lipidome result from “bulk” cells summing the heterogeneous nature in each tissue, and the results mainly reflect the lipid profiles of the major cell type or the major part of tissue. As for the brain, Arnold et al. reported similar results [42] when they examined cerebral cortex upon dietary EPA and DHA intake; the lipid changes in cerebral cortex were compared with other tissues including liver, kidney, heart, lung, and pancreas. Since cerebral cortex is the largest part of the brain, our whole brain data would have been significantly by the cerebral cortex. This was also the case for other tissues such as liver, white adipose, and skeletal muscle. Therefore, the lipidomes of different cell populations in each tissue should be investigated to further understand the mechanism of PUFA dietary intake in more detail.

ARA, EPA, and DHA supplementation induced various changes in lipid profiles. Dietary PUFA intake substantially increased the levels of free PUFAs and oxylipins, as well as the incorporation of PUFAs in phospholipids and triglycerides. In contrast, this side-by-side study revealed the unique features of ARA, EPA, and DHA metabolism. The basal amount of EPA was much lower than that of ARA and DHA in the plasma and tissues. This may have been because EPA easily undergoes degradation and/or conversion in tissues [43,44]. Dietary supplementation of EPA for 2 weeks elevated tissue EPA levels, and the profile of EPA-derived oxylipins was similar to that of ARA-derived oxylipins. However, the profile of DHA-derived oxylipins was a little bit different from those of ARA and EPA. These results may suggest that EPA and DHA have different metabolic compartments, which may be associated with their difference in biological effects.

In addition, dietary DHA was selectively incorporated into a unique phospholipid BMP. BMP is known as an acyl chain positional isomer of phosphatidylglycerol (PG), in which one acyl moiety is incorporated into the glycerol polar head, and the structures can be distinguished by positive ion mode MS/MS spectra, while the MS/MS spectra are the same in negative ion mode (Figure 5c). The biosynthetic and degradation pathways of BMP are still unclear; however, several enzymes, such as AGPAT8 and α/β hydrolase domain-containing 6 (ABHD6), have been reported as candidate enzymes involved in these pathways [40,41,45,46]. The subcellular localization of BMP is in the multivesicular membranes of late endosomes, where it enhances sphingolipid activator protein activity, resulting in (glyco) sphingolipid degradation [47,48,49]. In addition, BMP lipids contribute to membrane deformation, fusion, and transportation, as well as the incorporation of proteins and lipids [50], and one study reported that DHA-rich BMP is prone to peroxidation to prevent cholesterol oxidation [51]. However, the biological significance of DHA-BMP in endosome function and regulation is not well understood. 

PCA analysis (Figure 2) and circus plots (Figure 4 and Appendix A) displayed the metabolic fingerprints of different tissues. In the immune organs, the spleen, small intestine, and lung, high levels of PGs, TXs and fatty acid hydroxides were detected. In contrast, PUFA epoxides were the major metabolites in plasma and other tissues. CYPs are responsible for fatty acid epoxidation, and ARA-, EPA-, and DHA-derived epoxides have various actions, such as vasodilation, anti-inflammation, anticoagulation, and anti-allergy activity [23,24,25,26,27]. 

The comprehensive lipidomics approach revealed many findings about dietary PUFA metabolism. Our untargeted/targeted lipidomics data incorporating 1026 unique lipid molecules (Appendix A) are powerful when combined with other omics layers such as proteomics and transcriptomics. Recently, the lipidomics standards initiative (LSI) proposed three types of quantification in mass-spectrometry-based lipidomics: level 1, matching internal standards (IS) together with consideration of species-specific analytical response (essentially stating that stable-isotope-labeled lipids are preferred); level 2, matching IS where the lipid class between analyte and internal standard is identical; and level 3, non-matching IS where analytes are normalized with other lipid class molecules [52]. Although our untargeted lipidomics data have been displayed as the relative quantification in each tissue, our lipidome table can be normalized as level 3 and partially level 2 quantifications by using the internal standards. A consortium with the advances in analytical chemistry and informatics research would contribute to facilitating open data sciences through the reanalysis of deposited data [15].

This experimental design to investigate the PUFA metabolisms is expandable for further nutrient researches, and we believe that this study is a fundamental step towards clarification of the effects of dietary PUFA intakes in mice. Our lipidomics study offers a comprehensive picture of dietary PUFA metabolism in different tissues, and could provide an opportunity for data-driven hypotheses and biological insights into the molecular mechanisms of how different PUFA balances affect human and livestock health and disease. 

## Figures and Tables

**Figure 1 metabolites-09-00241-f001:**
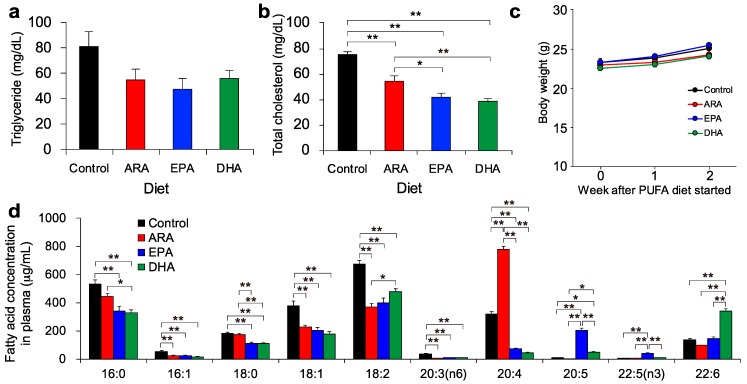
Effects on mouse plasma and body tissues following dietary supplementation with polyunsaturated fatty acids (PUFAs). (**a**) The total triglyceride and (**b**) total cholesterol in plasma were analyzed after two weeks of each PUFA diet. (**c**) The body weight in each PUFA diet condition was traced for two weeks, where the intake was started at 11 weeks of age. (**d**) The concentrations of total fatty acids in plasma were analyzed. The abbreviations ARA, EPA, and DHA, refer to arachidonic acid, eicosapentaenoic acid, and docosahexaenoic acid, respectively. The statistical significance was evaluated by Tukey’s testing (* *P* < 0.05 and ** *P* < 0.01) where one-way ANOVA (analysis of variance) showed the statistical significance among groups (*P* < 0.05).

**Figure 2 metabolites-09-00241-f002:**
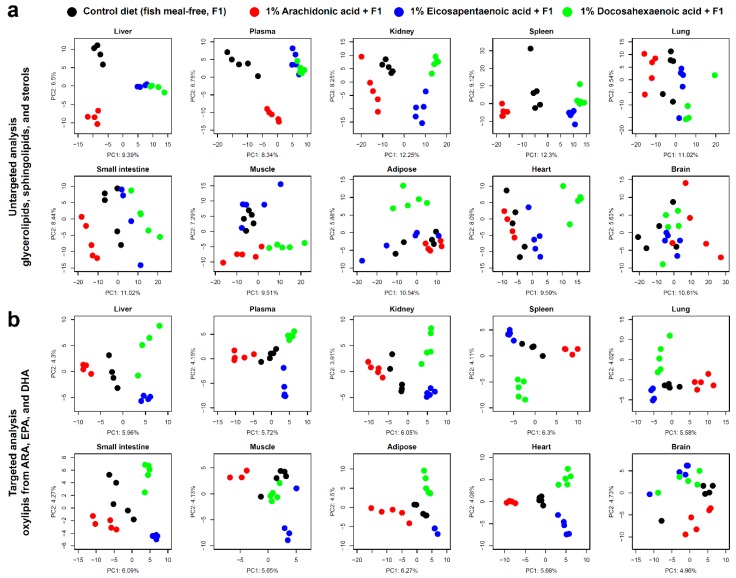
Principal component analyses (PCA) in untargeted and targeted lipidomics data. (**a**) The first and second principal components of PCA were described for plasma and all tissue metabolites obtained in the untargeted analysis. (**b**) The first and second components were also shown by the targeted lipidomics data. F1: fish-meal-free diet not containing ARA, EPA, or DHA.

**Figure 3 metabolites-09-00241-f003:**
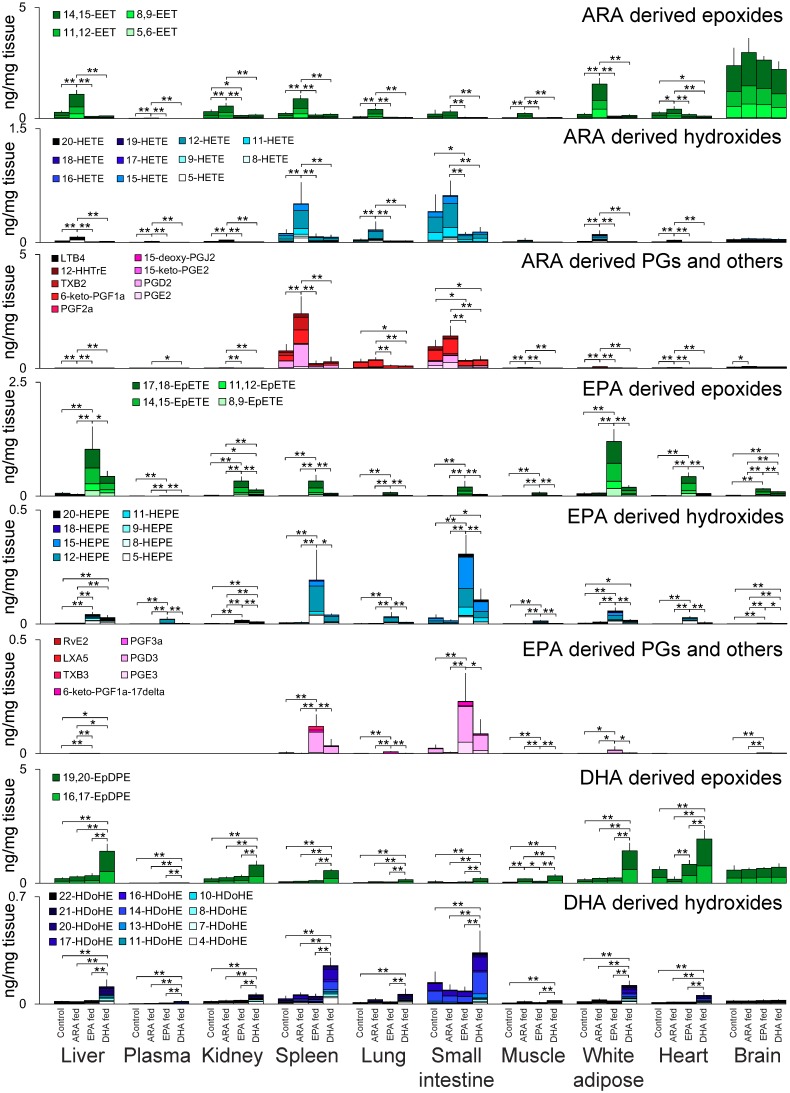
Oxylipin profiles in mouse tissues under each dietary condition. The total amounts of monoepoxides, hydroxides, and cyclized oxidized fatty acids including prostaglandins derived from ARA, EPA, and DHA are described. The results are given for each tissue in the four groups, i.e., the control, ARA-fed, EPE-fed, and DHA-fed groups. The ratio of oxidized fatty acids in each metabolite group is described by the stacked bar chart. The error bar indicates the standard deviation from the total amount values, and the statistical significance was evaluated by Tukey’s testing (* *P* < 0.05 and ** *P* < 0.01) where one-way ANOVA showed the statistical significance among groups (*P* < 0.05). For plasma, the y-axis unit becomes ng/µL plasma instead of ng/mg tissue.

**Figure 4 metabolites-09-00241-f004:**
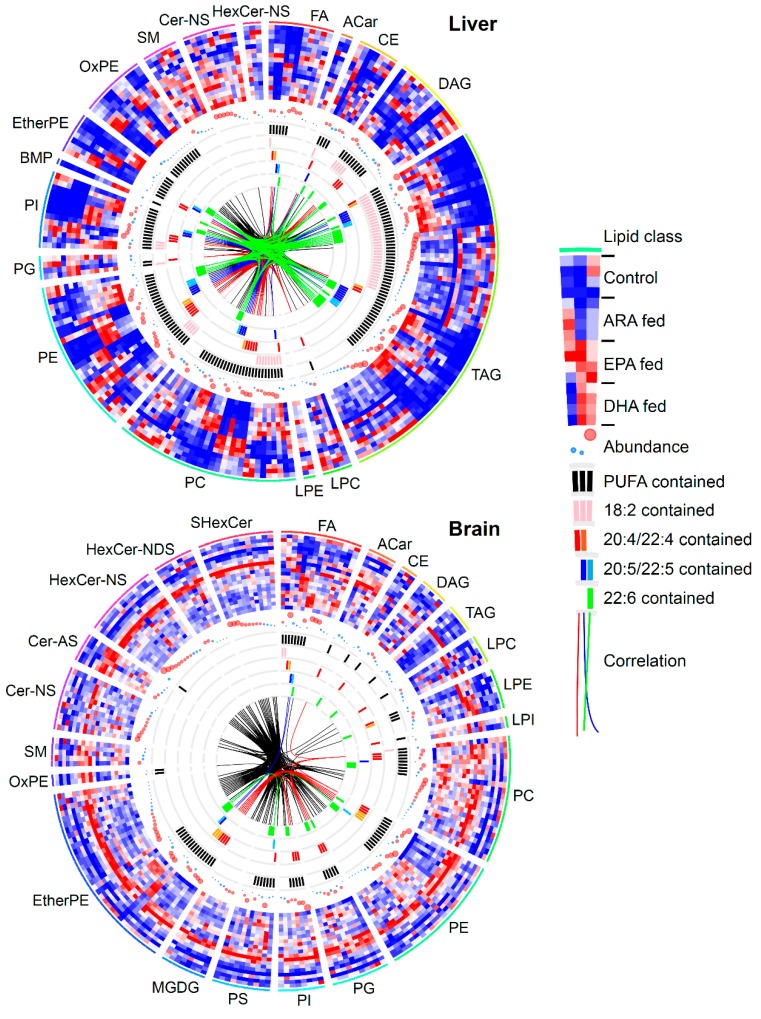
Lipid profiles in the liver and brain from the untargeted lipidomics data. In the circus plot, the lipid profiles are described separately by lipid class. The profile has been scaled in each metabolite from −1 to 1 in the heatmap. The ion abundance of each lipid is described by the circle size, where the average value in all conditions was utilized for the ion abundance size. Any lipid with more than two double bonds in an acyl chain was defined as a PUFA-containing metabolite. Lipids containing 18:2, 20:4, 22:4, 20:5, 22:5, and 22:6 are marked by pink, red, orange, blue, sky blue, and green colors, respectively. Metabolites were linked if the ion abundance correlation of two metabolites in biological samples was greater than 0.9.

**Figure 5 metabolites-09-00241-f005:**
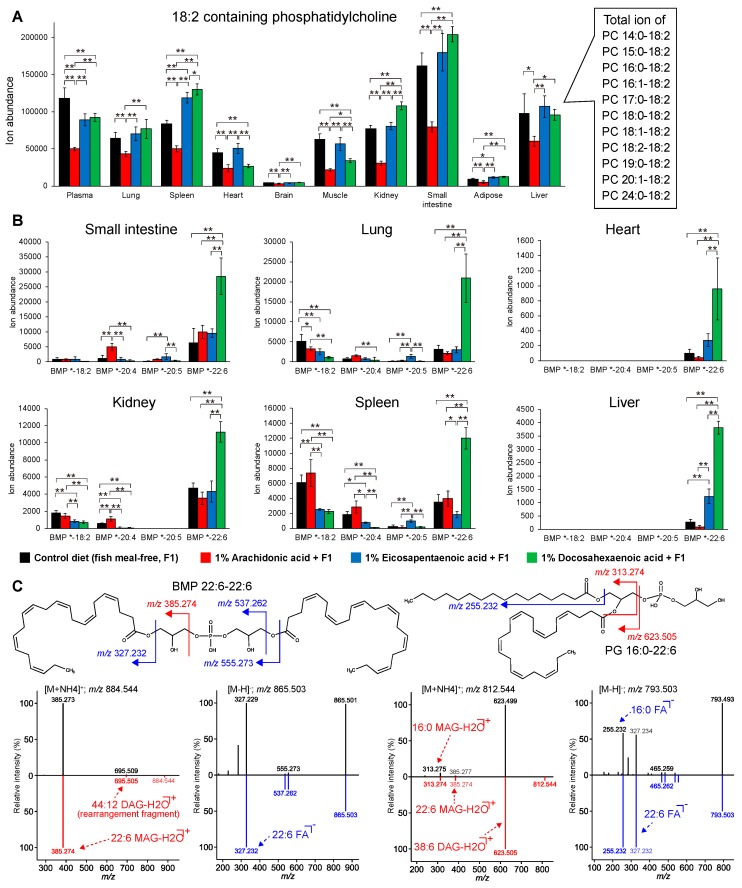
Unique lipid modulations in ARA or DHA dietary intakes. (**a**) The profile of phosphatidylcholines containing linoleic acid (18:2) as the longest PUFA chain and the ion abundances of each molecule. Note that a hyphen has been used for the description of acyl chains, e.g., 18:0-18:2, because the *sn1*/*sn2* acyl chain structural isomers were not distinguished in this study. (**b**) The profile of bis(monoacylglycero)phosphate (BMP): the asterisk (*) means any of the fatty acids and the ion abundances of lipids containing 18:2, 20:4, 20:5, or 22:6 were summed for quantification. Black, red, blue, and green represent the dietary conditions: control, ARA, EPA, and DHA supplementation, respectively. The statistical significance was evaluated by Tukey’s testing (* *P* < 0.05 and ** *P* < 0.01) where one-way ANOVA showed the statistical significance among groups (*P* < 0.05). (**c**) The experimental positive and negative ion mode MS/MS spectra (black color) of BMP 22:6-22:6 and PG 16:0-22:6 with the in silico reference spectrum (red color for ESI(+)-MS/MS and blue color for ESI(−)-MS/MS). The cleaved places generating the major fragment ions are described by red color for ESI(+)-MS/MS and blue color for ESI(−)-MS/MS. The product ion of *m/z* 695.5 from BMP 22:6-22:6 in positive ion mode was interpreted as the result of structural rearrangement in the fragmentation process.

**Table 1 metabolites-09-00241-t001:** Summary of characterized lipids in untargeted and targeted lipidomics.

Method	Class	Liver	Plasma	Kidney	Spleen	Lung	Small Intestine	Muscle	Adipose	Heart	Brain
Untargeted	FA	12	16	18	17	15	22	15	14	13	14
Untargeted	ACar	2	2	9	7	10	7	14	0	12	6
Untargeted	MAG	1	2	2	2	3	4	2	1	0	3
Untargeted	DAG	14	8	14	15	18	29	13	28	6	5
Untargeted	TAG	70	80	73	88	116	96	80	156	116	6
Untargeted	CE	7	10	4	6	2	5	3	0	3	1
Untargeted	LPC	7	13	8	7	8	7	7	1	7	6
Untargeted	LPE	4	6	10	9	8	8	6	0	4	7
Untargeted	LPG	0	0	0	1	1	0	0	0	0	0
Untargeted	LPS	0	0	1	0	0	1	0	0	0	0
Untargeted	LPI	2	0	3	1	0	2	0	0	1	4
Untargeted	PC	33	30	25	43	45	38	47	6	33	22
Untargeted	EtherPC	0	1	18	17	9	3	4	0	7	2
Untargeted	PE	27	9	34	29	32	28	31	4	28	25
Untargeted	EtherPE	8	3	24	40	35	26	16	2	21	35
Untargeted	PG	5	1	13	23	22	9	7	1	8	11
Untargeted	BMP	1	0	3	6	7	7	0	0	1	0
Untargeted	PS	2	0	6	20	11	2	1	0	1	10
Untargeted	PI	15	14	18	25	18	16	15	1	12	10
Untargeted	OxPC	2	0	9	0	0	5	5	0	8	0
Untargeted	OxPE	13	0	29	11	2	16	10	0	18	2
Untargeted	OxPG	0	0	0	0	0	0	0	0	0	0
Untargeted	OxPI	2	0	5	0	0	2	1	0	0	0
Untargeted	OxPS	0	0	7	2	2	1	0	0	1	0
Untargeted	Cer-NS	10	6	19	16	24	12	12	6	13	12
Untargeted	Cer-NDS	0	0	1	2	1	4	1	0	0	1
Untargeted	Cer-NP	0	0	4	0	4	8	0	0	0	1
Untargeted	Cer-AS	0	0	2	0	6	0	2	0	0	5
Untargeted	Cer-AP	0	0	0	0	1	5	0	0	0	0
Untargeted	HexCer-NS	4	4	8	11	12	5	4	2	2	13
Untargeted	HexCer-AP	0	0	4	0	0	15	0	0	0	0
Untargeted	SHexCer	0	0	5	0	2	0	2	0	0	14
Untargeted	SM	6	6	13	13	18	11	6	2	9	5
Targeted	FFA	17	17	17	16	17	17	17	17	17	16
Targeted	LA-O	7	7	5	6	6	7	7	7	7	7
Targeted	ALA-O	3	3	3	3	3	3	3	3	2	2
Targeted	GLA-O	1	0	0	0	0	1	1	0	0	0
Targeted	DGLA-O	1	2	1	2	2	4	4	3	1	1
Targeted	MA-O	1	2	1	2	0	2	2	1	1	1
Targeted	ARA-O	26	23	28	27	30	29	27	30	28	27
Targeted	EPA-O	16	13	13	15	14	19	13	19	14	13
Targeted	DHA-O	15	17	15	16	16	17	18	16	15	15

The nomenclature of lipid classes in untargeted analysis is described in RIKEN PRIMe website (http://prime.psc.riken.jp/). FFA: free fatty acid; LA-O, ALA-O, GLA-O, DGLA-O, MA-O, ARA-O, EPA-O, and DHA-O: oxylipins derived from linoleic acid, alpha-linolenic acid, gamma-linolenic acid, dihomo-gamma-linolenic acid, mead acid, arachidonic acid, eicosapentaenoic acid, and docosahexaenoic acid, respectively.

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
