# Peer review of "Characterization of Lipid Profiles after Dietary Intake of Polyunsaturated Fatty Acids Using Integrated Untargeted and Targeted Lipidomics"

_metabolites, 2019, doi:10.3390/metabo9100241_

Round 1
Reviewer 1 Report
In the manuscript “Characterization of lipid profiles after dietary intake of polyunsaturated fatty acids using integrated untargeted and targeted lipidomics”, Naoe et al. showed changes to the lipidome of C57BL/6 mice in response to diets containing ARA, EPA, or DHA. Specific comments are as follows:
1) How many mice were used per group? The supplementary data suggest that there are four per group, but this information was not provided anywhere in the main manuscript text; it needs to be included.
2) All of the data within Figures 1, 3, and 5; plus, Supplementary Figures 1 and 2; requires statistical analyses.
3) Under section 2.4, indicate the amount used for each internal standard.
4) Provide (as part of supplemental data) all calibration curves used.
5) Under section 3.1, the authors assign 18:1 as oleic acid and 18:2 as linoleic acid. Based on the methods used, it is possible that other fatty acids representing 18:1 and 18:2 are present (i.e. 18:1n-7). Carefully rewrite.
6) Figure 3 is very unclear. What are the four sets of bars per tissue representing? In looking at Supplementary Table 3, it appears that each bar represents one mouse. If this is the case, there is great variation and it needs to be explained, along with why some oxylipins were not detected between animals in the same group. Error bars would be useful, but the graph will not permit it; instead, prepare tables of all these data and provide statistical analyses.
7) Figure 5 is missing a figure legend for panel C. As part of panel A, listed are PC species; did the authors verify that 18:2 is in the sn-2 position of each PC species listed? The described methods would not readily provide that information. For the structures shown in panel C, show where the fragmentation occurs on each.
Author Response
We appreciated your valuable comments and suggestions to improve our manuscript. Please see the attachment including our point-by-point responses. Our responses were written in blue letters, and the major changes in our manuscript were also highlighted in blue letters.

Reviewer 2 Report
Authors completed a feeding experiment with mice using pure long chain polyunsaturated fatty acids. They analyzed several tissues and plasma using lipidomic approaches to evaluate the impact of feeding on the lipid species present in the tissues. This experiment did not have a stated hypothesis or objective, but contributes a more comprehensive view of changes in lipid species in response to long chain polyunsaturated fatty acid supplementation.
On line 44, you state that the ‘the amount is balanced in each cell and tissue to maintain homeostasis’. There is no reference for this statement and I’m not sure what mechanism you’re referring to. Perhaps I misunderstand the statement. Can you elaborate or clarify this statement, in response or revision or both?
Please consider making some statement about the heterogeneous nature of tissues. Many cell types are present in some tissues and small changes in a small cell population might be diluted by a cell type in greater abundance. I’m thinking of Figure 4, brain. Although there were fewer obvious changes in brain, some lipids in a small cell populations might be very meaningful but difficult to detect.
Was the whole brain used in preparing lipids for analysis? Was it homogenized? Even with the reference citation, it might be helpful to indicate that.
Arachidonic acid is not listed in Suppl Table 1. Was it undetectable? I would include a row with ‘0.0’ or ‘n.d.’ if it was to indicate it was measured and not detected.
Authors use the word ‘significantly’ on line 177 and a few other places in the manuscript but without applying any statistical analysis to verify that the effect cannot be random chance and to account for biological variation. Similarly for body weight on line 180. Although completing analyses of the volume of lipid species data from the lipidomic analyses would be difficult, completing 1-way ANOVA on the data in Figure 1a and 1b should be easy and would add validity to the use of the word ‘significantly’ here and for 1d also.
Authors state on line 186 that fatty acids were balanced by the dietary intakes. What does this mean? Can you clarify ‘balanced’?
Figure 4 – Very interesting figure. Also very complicated to discern meaning. Adding some text in the results section to better guide the reader to the specific region of the figure being cited might be helpful.
Figure 5 – The legend doesn’t seem to include a description of panel c.
Author Response

(The authors gave the same response as above.)

Reviewer 3 Report
Review
Manuscript ID: metabolites-593181
The article by Naoe and collaborators is concerned with untargeted and targeted lipidomic analysis of different organs, tissues and bio-fluids of mice after exposure to different diets. The assessed diets consisted of one control diet (free of ARA, EPA and DHA), and three additional diets supplemented with ethyl esters (EE) of ARA (EE-ARA-diet), EPA (EE-EPA-diet) or DHA (EE-DHA-diet).
The article is interesting, especially considering the variety of samples that were analyzed. However, the authors did not elaborate on published research indicating the lack of agreement between the effects of synthetic EE (ARA, EPA and DHA) and dietary triglycerides (TAG). Several differences exist with regard to how the body handles EE versus TAG. Some studies have shown that EE are not absorbed in their manufactured synthetic form. For example, dietary interventions have concluded that EE-EPA and EE-DHA do not appear in the blood of humans and rats after the intake of large amounts of the EE forms (Hamazaki et al. Prostaglandins 1982; Terano et al. Atherosclerosis 1983; Hamazaki et al. Lipids 1987). It is important to elaborate about the potential mechanism of EE absorption in the different tissues, organs and bio-fluids with appropriate references.
Another aspect of concern is the poor data analysis. I have the impression that the authors are using sophisticated multivariate chemometrical techniques without having an idea of the basic principles behind them. For example, the authors performed a PCA analysis and show the score plots for the various organs in (Fig. 2). The explanation of Fig. 2 reads “The score plots of the principal component analysis (PCA) from untargeted lipidomics indicated that the glycerolipids and glycerophospholipids were clearly changed in the metabolic organs, such as the liver and kidney, while the lipidome was not significantly changed in the brain and muscle tissues, including the heart and skeletal muscle (Fig. 2a)”
Where in Fig.2 I can see the alleged changes? The authors are oblivious to the fact that a PCA score plot CANNOT show relationships between the measured variables (e.g glycerolipids, glycerophospholipids, etc). It is an impossible task. If in doubt, the authors can check the 20 different figures portrayed in Fig. 2 and see that the only relevant feature of these figures is the correlation between the four different diets (indicated by black, blue, green and red dots) in the different organs. The full discussion of Fig 2 is pure speculation. If the score plots are showing relationships between the various diets, how the authors can conclude matter of fact that “…the score plots of DHA dietary intake were classified independently from the other conditions in the skeletal muscle, adipose, and heart, which indicates that those tissues preferably incorporate DHA over ARA and EPA” I wonder where in the plot they could see the alleged incorporation. I suggest reporting the loading plots.
I read very carefully the explanation about Fig. 3 and also the circus plot (Fig. 4) and I cannot see all the features that the authors are describing in the text. Both figures are very confusing and hard to digest. By reading their captions, I cannot extract some rational information from these plots. Maybe the authors have not explained them properly or perhaps they are not the best option for displaying the results. For example, Fig 3a,b,c show four tiny sub-bars per sample, I assumed that they correspond to control, ARA, EPA and DHA. However, the caption of Fig. 3 indicates that Fig. 3a, Fig.3b and Fig. 3c correspond to ARA, EPA and DHA respectively. Then, what exactly are the tiny sub-bars in the figures? It is very confusing.
Author Response

(The authors gave the same response as above.)

Round 2
Reviewer 1 Report
All previous comments were addressed. The use of Bonferroni testing increases the chances of false negative outcomes. ANOVA coupled with Tukey's testing would be more stringent; please re-assess all statistical analyses with the latter.
Author Response
We addressed the reanalysis by using Turkey’s testing, and Figures were revised accordingly.
Reviewer 3 Report
My only suggestion is to make clear in the abstract and in the introduction that the diets were supplemented with the synthetic ethyl ester forms of ARA, EPA and DHA.
Author Response
Comment: My only suggestion is to make clear in the abstract and in the introduction that the diets were supplemented with the synthetic ethyl ester forms of ARA, EPA and DHA.
Response: We added the following sentences to describe the detail in the abstract and the introduction.
Abstract
Here, we performed lipidomic analyses on mouse plasma and nine tissues, including the liver, kidney, brain, white adipose, heart, lung, small intestine, skeletal muscle, and spleen, for the dietary intake condition of arachidonic acid (ARA), eicosapentaenoic acid (EPA), and docosahexaenoic acid (DHA) as the ethyl ester form.
Introduction
In this study, we investigated the lipid profiles in plasma and tissues including the liver, kidney, white adipose, skeletal muscle, heart, small intestine, lung, brain, and spleen after dietary intake of the ethyl ester form of ARA (ARA-E), EPA (EPA-E) or DHA (DHA-E) in mice.